# Inherited Retinal Dystrophy in Southeastern United States: Characterization of South Carolina Patients and Comparative Literature Review

**DOI:** 10.3390/genes13081490

**Published:** 2022-08-20

**Authors:** Joseph Griffith, Kareem Sioufi, Laurie Wilbanks, George N. Magrath, Emil A. T. Say, Michael J. Lyons, Meg Wilkes, Gurpur Shashidhar Pai, Mae Millicent Winfrey Peterseim

**Affiliations:** 1Department of Ophthalmology, Medical University of South Carolina, Charleston, SC 29425, USA; 2Greenwood Genetic Center, Greenwood, SC 29646, USA; 3Department of Genetics, Medical University of South Carolina, Charleston, SC 29425, USA

**Keywords:** retinal dystrophy, retinitis pigmentosa, Stargardt disease, Usher syndrome, Southeastern United States

## Abstract

Inherited retinal dystrophies (IRDs) are a group of rare diseases involving more than 340 genes and a variety of clinical phenotypes that lead to significant visual impairment. The aim of this study is to evaluate the rates and genetic characteristics of IRDs in the southeastern region of the United States (US). A retrospective chart review was performed on 325 patients with a clinical diagnosis of retinal dystrophy. Data including presenting symptoms, visual acuity, retinal exam findings, imaging findings, and genetic test results were compiled and compared to national and international IRD cohorts. The known ethnic groups included White (64%), African American or Black (30%), Hispanic (3%), and Asian (2%). The most prevalent dystrophies identified clinically were non-syndromic retinitis pigmentosa (29.8%), Stargardt disease (8.3%), Usher syndrome (8.3%), cone-rod dystrophy (8.0%), cone dystrophy (4.9%), and Leber congenital amaurosis (4.3%). Of the 101 patients (31.1%) with genetic testing, 54 (53.5%) had causative genetic variants identified. The most common pathogenic genetic variants were *USH2A* (n = 11), *ABCA4* (n = 8), *CLN3* (n = 7), and *CEP290* (n = 3). Our study provides initial information characterizing IRDs within the diverse population of the southeastern US, which differs from national and international genetic and diagnostic trends with a relatively high proportion of retinitis pigmentosa in our African American or Black population and a relatively high frequency of *USH2A* pathogenic variants.

## 1. Introduction

Inherited retinal dystrophies (IRDs) are rare diseases defined by specific clinical and molecular features leading to significant visual impairment. In high- and middle-income countries, IRDs are the third most common cause of childhood blindness [1]. IRDs represent a heterogenous group of disorders that are characterized by progressive retinal dysfunction or degeneration. The largest subgroup of IRDs are the pigmentary retinopathies, whose pathology is related to inappropriate development or loss of the retinal photoreceptors [2]. The most common form of IRD is non-syndromic retinitis pigmentosa (RP), which is a predominately rod-cone degeneration.

The prevalence of each form of IRD varies widely depending upon the patient population and geographic area [3,4,5]. The worldwide prevalence of IRDs is estimated to be 1/2000 [6]. RP is found in 1/4000 people in developed countries but can reach up to 1/230 in populations with high rates of consanguinity [4,7,8,9,10,11,12]. Subsequently, the relative proportion of each form of IRD also varies widely, and many of the cited reports on prevalence are extrapolated from older, smaller studies [13,14,15,16,17,18]. Because IRDs are characterized by genetic and clinical heterogeneity, gene identification and mutation analysis are challenging but important. To date, there are 340 identified genes that cause IRDs and are catalogued by RetNet [19]. Of those mutations, 10 genes are responsible for 68% of IRD cases [20]. However, the genetic cause remains unknown in 28–47% of patients with an IRD who undergo genetic testing [21,22,23,24,25,26]. Thus, there is a need to further characterize the genetic landscape of IRDs to gain a better understanding of these dystrophies. Additionally, it is of great importance to publish prevalence data about previously known IRD-causing genetic mutations to direct investigations of targeted gene therapy.

The majority of the patients in this study live in South Carolina (SC), which is located in the Southeastern US and has an ethnically and culturally diverse population of over 5.1 million [27]. The population is largely comprised of White and African American or Black residents, at 68.6% and 27.0%, respectively. This proportion of African American or Black residents is much higher than the national average of 14.2% and is representative of US southeast and midatlantic coastal states [27]. This study reports the frequency of each type of reported IRD and the characteristics of successful genetic variant identification. These results serve as a basis for future investigation toward understanding and developing treatment for these diseases.

## 2. Materials and Methods

Institutional review board approval was obtained, and the study was conducted in accordance with the Declaration of Helsinki and the International Conference on Harmonization. The medical records of patients from the Medical University of South Carolina and the Greenwood Genetic Center over a 9-year period from April 2013 to February 2021 were retrospectively reviewed. Patients with the following diagnoses were included in this study: Retinitis Pigmentosa, Leber congenital amaurosis (LCA), Bardet Biedl syndrome, Usher syndrome, Stargardt disease, Retinal degeneration (peripheral), unspecified hereditary retinal dystrophy, Dystrophy cone (progressive), Dystrophy rod (progressive), and hereditary fleck dystrophy. Patients with multiple listed diagnoses were included with the most specific diagnosis available. Data including demographics, presenting symptoms, visual acuity, retinal exam findings, imaging findings, and genetic test results were gathered.

Because this study is a retrospective chart review, patients in this cohort did not undergo systematic genetic testing, and a wide variety of genetic tests were performed based on the clinical judgement of the treating physician. A list of all the known genetic tests performed in our cohort is listed by diagnosis in the supplemental information (Appendix A). Many of the tests reported gene variants with undetermined clinical significance. For completeness, all of the genetic variations found in this study’s population, regardless of previously known clinical significance, are reported in the Appendix A. Only mutations that were deemed pathogenic or likely pathogenic were included in the genetic analysis. Patients with multiple diagnoses, such as RP and Usher syndrome, were analyzed by their most specific diagnosis, which would be Usher syndrome in this example.

The relative proportions of IRDs were compared to the largest American genetic ophthalmologic study to date, the eyeGENE Network, which includes over 6000 patients and is considered in this study to represent the broader US [20]. The relative proportions of the largest disease entities and pathogenic variants were also compared to various international IRD cohorts.

## 3. Results

### 3.1. Demographics

There were 325 patients included in this study. The mean age was 37.8 years (median 35, range: 1 to 81 years). The race of 15.1% of the patients was unknown or unspecified. Of the patients with known race, 64.1% were White, 29.7% were African American or Black, 3.3% were Hispanic, 2.2% were Asian, and 0.7% were multi- or bi-racial. Patients originated from across SC, with 25.5% of the patients from Charleston county and 7 from nearby states, including North Carolina and Georgia.

### 3.2. Distribution of IRDs Based on Clinical Diagnosis

The most common clinical diagnosis was non-syndromic RP (29.8%), followed by Stargardt disease (8.3%), Usher syndrome (8.3%), and Cone-rod dystrophy (8.0%) (Figure 1). Patients with non-specific diagnoses were grouped together and comprised 8.3% of the study population. There were 32 unique diagnoses in the cohort of 325 patients, and 27.4% reported positive family history. There were a number of syndromic forms of RP in the cohort, including Usher syndrome (8.3%), Batten disease (2.8%), and Bardet–Biedl syndrome (0.9%). The group “Others” comprises specific disease entities that were only found once in the cohort, which includes Aicardi syndrome, Alstrom syndrome, Cobalamine C syndrome, Cohen syndrome, Donye honeycomb macular dystrophy, lattice degeneration, Lowry Wood syndrome, Marshal syndrome, Noonan syndrome, pattern dystrophy, Optiz B/GGG syndrome, and Zellweger syndrome (Figure 1).

### 3.3. Genetic Findings

Genetic testing was performed in 100 patients (31.1%). The majority (70%) of patients with testing were White, while only 17% were Black or African American. Of the patients with testing, 54 patients (53.5%) obtained a positive test, identifying one or more variants that were determined to be causative. The gene with the most pathogenic variants identified in the cohort was *USH2A* in 11 patients (Table 1, Figure 2), with 10 being diagnosed clinically with Usher syndrome. *ABCA4* pathogenic variants were found in eight patients, all of whom were diagnosed as Stargardt disease. *CLN3* pathogenic variants were found in seven patients, all of whom were diagnosed with Batten disease. *CEP290* mutations were found in three patients, all of whom were diagnosed with LCA (Table 1, Figure 2). All patients with testing data, whether confirmed pathogenic or unknown, are listed with the diagnosis, specific nucleotide and/or amino acid alterations, gene, allele state, clinical significance, and disease inheritance (Appendix A).

### 3.4. Disease Characteristics

The characteristics of each disease diagnosis with more than 14 patients are compiled in Table 2. In patients with RP, the age of diagnosis varied greatly, with 15% of the patients being diagnosed in the first decade of life. Other patients with RP presented symptomatically in their twenties, thirties, and forties, with the latest onset of symptoms at age 58. The most common presenting symptom was nyctalopia (71%). The most common retinal finding was bony spicules (69%). The genetic findings associated with RP were heterogenous, with no pathogenic variants seen more than twice (Table 1). Almost half (44%) of the RP patients identified as Black or African American, larger than the proportion of Black or African American in this study’s IRD cohort (30%) and in SC (27%) [27].

Patients diagnosed with Usher syndrome (n = 27) represented the largest group of syndromic RP patients. The onset of visual symptoms was preceded by hearing loss, which occurred before the age of 4 in 100% of the patients. Many (37%) of the Usher syndrome patients were diagnosed in the first decade of life due to congenital hearing loss. The most common presenting visual symptoms were blurriness or nyctalopia and began as early as the age of 8 and as late as 55. Most (90%) of the patients developed visual symptoms by their early thirties. As with non-syndromic RP, the most common retinal finding was bony spicules (56%) (Table 2). The majority (65%) of the Usher syndrome population with genetic testing was found to have a pathogenic variant in *USH2A* (Table 1).

In patients with the clinical diagnosis of Stargardt disease (n = 27), the onset of visual loss began between the ages of 7 and 40, with most patients (83%) having visual difficulty prior to the age of 30. The proportion of patients with Stargardt disease who identified as white (77%) was larger than that of this study’s IRD cohort (64%) and of SC (64%) (Table 2). The two most common retinal findings included macular atrophy (48%) and flecks (32%) (Table 2). All of the patients with Stargardt disease that had positive genetic testing demonstrated pathogenic variants in the *ABCA4* gene (Table 1).

In patients with the clinical diagnosis of Cone-rod dystrophy (n = 26), the onset of visual impairment began from birth to after the age of 40. However, most patients reported having visual difficulties in childhood, with only 23% of patients presenting after the age of 20. The two most common retinal findings were pigmentary changes (30%) and bone spicules (22%) (Table 2). The genetic findings for cone-rod dystrophy were heterogenous, with no mutation being found more than once (Table 1).

### 3.5. National and International Comparisons

The most common genetic variants and diseases from IRD cohorts in the US and internationally were compared to those from this study’s cohort (Table 3 and Table 4) [9,20,24,26,28,29,30]. This cohort’s proportion of RP is comparable to that of the eyeGENE findings, which represents the broader US IRD population, at 29.8% and 38.4%, respectively (Table 3). Our Stargardt disease proportion (8.3%) was about a third of the eyeGENE’s proportion (24.0%), while the proportion of Usher syndrome (8.3%) was greater than double that of the eyeGENE network (3.7%). In our cohort, the LCA proportion (4.3%) was over four-fold more common than in eyeGENE’s cohort (0.9%).

Overall, our Southeastern US cohort of IRDs was broadly similar to those of international cohorts, with RP being the dominant IRD (Table 4). The second and third most common diseases found in the cohorts included Stargardt disease, Cone-rod/Rod-cone dystrophy, and Usher Syndrome in various orders (Table 4). The most common pathogenic variants found in the broader US and international IRD cohorts, with the exception of China, were *ABCA4* followed by *USH2A*, while in our cohort it was *USH2A* followed by *ABCA4*.

## 4. Discussion

We found a relatively high proportion of Usher syndrome and relatively low population of Stargardt disease in the SC population, compared to the broader US and international IRD populations [9,20,24,26,28,29,30] (Table 3 and Table 4). SC’s Usher syndrome proportion was greater than double that of the broader US, while SC’s Stargardt disease proportion was just a third of the broader US proportion [20].

The SC IRD cohort contains a substantially larger proportion of African American or Black patients at 30% of the population with known race compared to 9% in the eyeGENE cohort, which represents the broader US [20,27]. The higher proportion of African American or Black patients in our cohort reflects the racial demographics of the Southeastern US compared to the broader US. Differences in the racial demographics of the individual diseases may also explain some of the relative differences in the disease proportions of the SC and broader US cohort. For example, the Stargardt disease population of the SC cohort was majority White (77%), which could explain the relative decrease in the Stargardt disease proportion compared to the broader US, given that the SC population contained far less White patients proportionally than the broader US cohort. In addition, our Black or African American IRD cohort demonstrated a relatively high percentage of RP (44%).

An important limitation of this study was that genetic testing was not performed routinely in this cohort of IRD patients. Only 31.1% of our cohort received genetic testing, and the tests offered were not homogenous. However, similar testing methods have been used in comparable studies [20,24,26]. Heterogeneity in genetic testing resulted from variations in testing availability, including insurance coverage, and from provider and patient opinion of testing importance. In the majority of the patients, testing was offered but declined due to severity of disease and/or lack of available treatment. Another study limitation is that the available testing, even within the brief 9-year period during which patient data were collected, has expanded dramatically as new gene panels with next-generation sequencing are added to the repertoire. While today pathogenic mutations in over 340 genes are known to cause IRDs, in 2013, there were only 175 known genes causing IRDs, which is up from just 60 in 2003 [9,19]. It has been suggested that increased access to molecular tests could be accomplished through the development of smaller and cheaper gene panels according to the IRD genetic profile of each population [24]. A final limitation of this study is the ambiguity of some of the clinical diagnoses given the diverse presentation of phenotypes and the absence of genetic testing for many patients.

## 5. Conclusions

IRDs represent a broad group of heterogeneous diseases that lead to loss of vision. We provide initial information characterizing IRDs within the diverse population of Southeastern US, noting differences from broader US and international IRD population studies, including a relatively high proportion of RP in our Black or African American population as well as the relatively high frequency of the *USH2A* gene variants. Increased knowledge of disease characteristics and genetic analysis of diverse populations will better enable clinicians and researchers to target therapies and future research.

## Figures and Tables

**Figure 1 genes-13-01490-f001:**
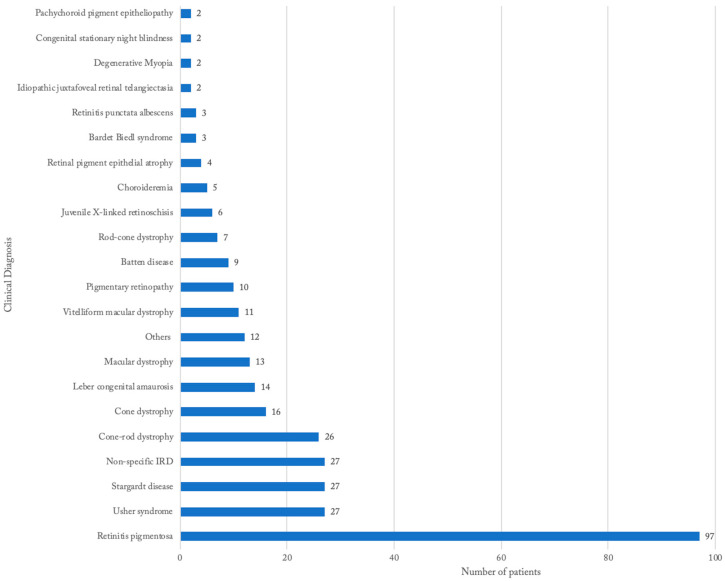
Distribution of clinical diagnosis among South Carolina patients with IRD.

**Figure 2 genes-13-01490-f002:**
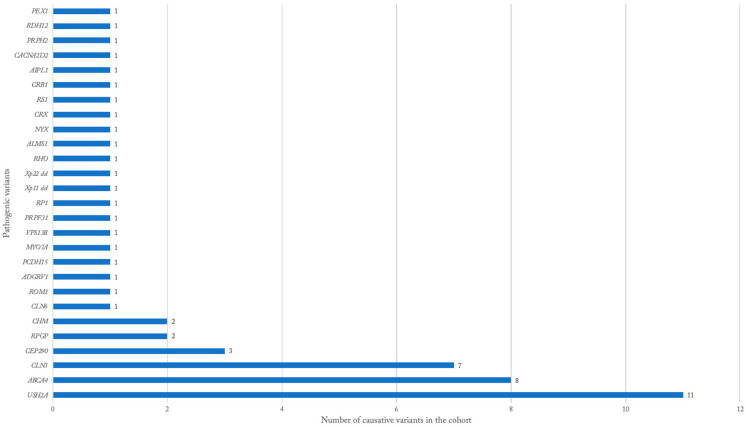
Distribution of the pathogenic genetic results.

**Table 1 genes-13-01490-t001:** Pathogenic genetic variants per retinopathy identified in our cohort.

Disease	Confirmed Disease-Causing Genes (n)
Alstrom cone dystrophy	*ALMS1 (1)*
Batten disease	*CLN3 (7), CLN6 (1)*
Choroideremia	*CHM (2)*
Cohen syndrome	*VPS13B (1)*
Cone-rod dystrophy	*RPGR (1), USH2A (1)*
Congenital stationary night blindness	*NYX (1)*
Juvenile X-linked retinoschisis	*RS1 (1)*
Leber congenital amaurosis	*CEP-290 (3), AIPL1 (1), CACNA2D2 (1), CRB1 (1)*
Opitz G/BBB syndrome	*Xp22 deletion (1)*
Pigmentary retinopathy	*PRPH2 (1)*
Retinitis pigmentosa	*PRPF31 (1), RDH12 (1), RHO (1), RP1 (1), RPGR (1), RP1L1 (1), ROM1 (1), Xp11 del (1)*
Stargardt disease	*ABCA4 (8)*
Non-specific IRD	*CRX (1)*
Usher syndrome	*USH2A (10), ADGRV1 (1), MYO7A (1), PCDH15 (1)*
Zellweger Syndrome	*PEX1 (1)*

**Table 2 genes-13-01490-t002:** Individual disease characteristics.

Disease	n	Age Range	Visual Acuity Range	Positive Family History	Race	Retinal Findings	Other Ocular Pathologies
Retinitis pigmentosa	97	5 to 78	20/20 to NLP	31%	White (55%), Black (44%), Hispanic (1%)	bony spicules (69%), vascular attenuation (62%), optic nerve pallor (36%), atrophy (18%), pigment mottling/clumping (15%)	early cataract (16%), nystagmus (9%), strabismus (7%), glaucoma (7%), cystoid macular edema (4%), keratoconus (2%)
Usher syndrome	27	5 to 60	20/20 to HM	27%	White (62%), Black (29%), Hispanic (5%), Asian (5%)	bony spicules (56%), vascular attenuation (44%), optic nerve pallor (31%), normal (19%), atrophy (13%)	early cataract (7%), hyperopic astigmatism (7%) vitreous detachment (4%), glaucoma (4%), diplopia (4%), corneal clouding (4%), Sjogren’s syndrome (4%)
Stargardt disease	27	10 to 79	20/30 to 20/800	42%	White (77%), Black (23%)	macular atrophy (48%), flecks (32%), pigmentary changes (28%), bull’s eye (20%), beaten metal appearance (8%)	myopic astigmatism (8%), strabismus (4%), posterior vitreous detachment (4%), asteroid hyalosis (4%)
Cone-rod dystrophy	26	12 to 57	20/20 to LP	38%	White (60%), Black (30%), Hispanic (10%)	pigmentary changes (30%), bone spicules (22%), attenuation (22%), pallor (22%), RPE changes (13%), atrophy (13%), foveal hypopigmentation (9%), tapetal reflex (9%), normal (9%)	myopia (12%), strabismus (8%), nystagmus (8%)
Cone dystrophy	16	8 to 74	20/30 to 20/800	38%	White (64%), Black (29%), Hispanic (7%)	atrophy (43%), bull’s eye maculopathy (21%), normal (21%), vascular attenuation (14%)	nystagmus (13%), amblyopia (6%), high myopia (6%)
Leber congenital amaurosis	14	1 to 31	20/30 to NLP	30%	White (62%), Hispanic (14%), Black (8%), Asian (8%), Bi-racial (8%)	pigmentary changes (36%), pallor (27%), attenuation (18%), atrophy (18%)	nystagmus (86), high hyperopia (43%), strabismus (36%), oculo-digital sign (14%)

**Table 3 genes-13-01490-t003:** Relative proportions of IRDs in the eyeGENE report [20] compared to that of our southeastern US cohort.

Disease	Eye Gene Percentage	SC Percentage	Prevalence in the Literature [References]
Retinitis pigmentosa	38.4	29.8	1 in 3000–4000 [4,7,8,9,10,11,12]
Usher syndrome	3.7	8.3	1 in 6000–25,000 [13,31]
Stargardt disease	24.0	8.3	1 in 8000–10,000 [14]
Cone-rod dystrophy *	8.5	15.1	1 in 40,000 [32]
Leber congenital amaurosis	0.9	4.3	1 in 50,000–100,000 [15]
Best vitelliform macular dystrophy	4.0	3.4	1 in 16,500–21,000 [16,33,34]
Batten disease	0.0	2.8	1 in 25,000–50,000 [35]
X-linked juvenile retinschisis	3.3	1.8	1 in 5000–25,000 [17,18]
Choroideremia	4.3	1.5	1 in 50,000–100,000 [36]
Doyne Honeycomb dystrophy	1.7	0.3	unknown
Pattern dystrophy **	4.9	0.3	1 in 7400–8200 [33]
FEVR (Familial exudative vitreoretinopathy)	2.5	0	unknown
Bietti crystalline corneal–retinal dystrophy	0.5	0	1 in 100,000–135,000 [37]
Kearns–Sayre syndrome	0.1	0	unknown
Congenital stationary night blindness/Oguchi disease	1.4	0.6	unknown
Occult macular dystrophy	0.6	0	unknown
Stickler syndrome	0.4	0	1 in 7500–9000 [38]
Sorsby dystrophy	0.6	0	1 in 220,000 [39]

* Includes cone-rod dystrophy, rod-cone dystrophy, and cone dystrophy. ** Includes adult-onset foveomacular dystrophy in eyeGene study reference 20.

**Table 4 genes-13-01490-t004:** Various international IRD cohorts and their most common genetic and diagnostic findings.

Location [References]	Number of Patients	Year	1st Most Common Gene	2nd Most Common Gene	3rd Most Common Gene	1st Most Common Disease	2nd Most Common Disease	3rd Most Common Disease
SC	325	2022	*USH2A*	*ABCA4*	*CLN3*	RP	cone-rod/rod-cone dystrophy	Usher and Stargardt tied
Brazil [24]	1246	2018	*ABCA4*	*USH2A*	*CEP-290*	RP	Leber congential amaurosis	Stargardt disease
Israel [26]	2420	2020	*ABCA4*	*USH2A*	*FAM161A*	RP	Stargardt disease	cone-rod/rod-cone dystrophy
USA [20]	5385	2020	*ABCA4*	*USH2A*	*RPGR*	RP	Stargardt disease	cone-rod/rod-cone dystrophy
France [9]	1957	2013	*ABCA4*	*USH2A*	*MYO7A*	RP	Usher syndrome	cone-rod/rod-cone dystrophy
UK [28]	4236	2020	*ABCA4*	*USH2A*	*RPGR*	n/a	n/a	n/a
Germany [29]	2158	2020	*ABCA4*	*USH2A*	*RPGR*	RP	Macular Dystrophy	Cone-rod/rod-cone dystrophy
China [30]	319	2018	*USH2A*	*RPGR*	*CYP4V2*	RP	Cone-rod/rod-cone dystrophy	Usher syndrome

## Data Availability

Detailed genetic data can be found in the Appendix A. Further data requests can be made by contacting the corresponding author.

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
