# Peer review of "Inherited Retinal Dystrophy in Southeastern United States: Characterization of South Carolina Patients and Comparative Literature Review"

_genes, 2022, doi:10.3390/genes13081490_

Round 1

Reviewer 1 Report

Thank you for sharing this interesting work.There are a few concerns though,that need to be addressed.

1.Since the work has been submitted to gene,please mention in details about the genetic tests run on the cohort to diagnose the IRDs.There is no mention of these details in the materials and methods

2.The references are incomplete and not matching with the text.

3.A chinese study has been quoted in the section on comparison with the international and national data,it is missing in the reference section.

4.Please put reference number on all the international and national studies

Reviewer 2 Report

Title: Inherited Retinal Dystrophy in Southeastern United States: Characterization of South Carolina Patients and Comparative Literature Review

Authors: Joseph Parkwood Griffith III , Kareem Sioufi , Laurie Wilbanks , George Magrath , Emil Say , Michael J Lyons , Meg Wilkes , Gurpur Shashidhar Pai , Mae Millicent Winfrey Peterseim

The authors present a research study regarding Inherited Retinal Dystrophy (IRD) in the Southeastern United States, which can be seen as relevant to a broader scientific background. Overall, it is believed that this is an interesting but not unexpected (due to different racial backgrounds and already known prominent IRDs in various racial backgrounds) research summary of IRDs in the southeastern united states. The authors beautifully discuss and lay out all challenges and limitations of their research study in the discussion section, which is very much appreciated and addresses all questions raised.
